# Identification of a Novel Equine Papillomavirus in Semen from a Thoroughbred Stallion with a Penile Lesion

**DOI:** 10.3390/v11080713

**Published:** 2019-08-04

**Authors:** Ci-Xiu Li, Wei-Shan Chang, Katerina Mitsakos, James Rodger, Edward C. Holmes, Bernard J. Hudson

**Affiliations:** 1Marie Bashir Institute for Infectious Diseases and Biosecurity, Charles Perkins Centre, School of Life & Environmental Sciences and Sydney Medical School, The University of Sydney, Sydney, NSW 2006, Australia; 2Royal North Shore Hospital, Clinical Microbiology & Infectious Diseases, Reserve Road, St Leonards, NSW 2065, Australia; 3Vets & Veterinary Surgeons, Jerry Plains Veterinary Hospital, 10 Pagan Street, Jerry Plains, NSW 2330, Australia

**Keywords:** equine papillomaviruses, horse, genital wart, phylogeny, evolution

## Abstract

Papillomaviruses (PVs) have been identified in a wide range of animal species and are associated with a variety of disease syndromes including classical papillomatosis, aural plaques, and genital papillomas. In horses, 13 PVs have been described to date, falling into six genera. Using total RNA sequencing (meta-transcriptomics) we identified a novel equine papillomavirus in semen taken from a thoroughbred stallion suffering a genital lesion, which was confirmed by nested RT-PCR. We designate this novel virus *Equus caballus papillomavirus 9* (EcPV9). The complete 7656 bp genome of EcPV9 exhibited similar characteristics to those of other horse papillomaviruses. Phylogenetic analysis based on concatenated E1-E2-L2-L1 amino acid sequences revealed that EcPV9 clustered with EcPV2, EcPV4, and EcPV5, although was distinct enough to represent a new viral species within the genus *Dyoiotapapillomavirus* (69.35%, 59.25%, and 58.00% nucleotide similarity to EcPV2, EcPV4, and EcPV5, respectively). In sum, we demonstrate the presence of a novel equine papillomavirus for which more detailed studies of disease association are merited.

## 1. Introduction

Equine papillomaviruses are small, non-enveloped viruses that comprise a circular and double-stranded DNA genome of up to approximately 8 kb in length. These viruses are associated with a variety of equine diseases including classical viral papillomatosis, genital papillomatosis, and aural and genital plaques. To date, 13 species of papillomavirus have been documented to infect the Equidae (horses and donkeys): *Bos taurus papillomavirus 1* (BPV1) [1], *2* (BPV2) [1], and *13* (BPV13) [2], *Equus caballus papillomaviruses 1–8* (EcPV1–8) [3,4,5,6,7,8], and *Equus asinus papillomaviruses 1–2* (EaPV1–2) [9]. These viruses are further classified into six genera based on the level of nucleotide sequence diversity in the L1 gene. Notably, these viruses have been isolated from a variety of lesions (aural plaques, genital masses, verrucous) and almost all seem to be associated with distinct pathologies (Appendix A). Herein, we describe the detection of a novel papillomavirus in a 12-year-old thoroughbred Australian stallion using bulk RNA sequencing (meta-transcriptomics).

During the 2018 southern hemisphere serving season, the stallion experienced difficulty covering mares, primarily manifest as apparent pain on ejaculation. A wart-like lesion, 1 cm in circumference, was observed at the tip of the penis consistent with a genital papillomavirus lesion (Appendix A). Further endoscopy and ultrasound excluded neoplasia with no evidence of further internal lesions.

## 2. Materials and Methods

### Sample Collection, RNA Sequencing and Virus Discovery

Two sets of urine and semen samples were collected from the stallion for microbiological investigation, one set placed into a standard specimen container and the other stored in RNA later. Because the nature of any causative pathogen was unknown, we employed a meta-transcriptomic approach as this is able to detect any microbial species (i.e., bacteria, eukaryotes, viruses) as long as sufficient expressed RNA is present. Accordingly, total RNA was extracted using the RNeasy plus universal kit (QIAGEN, Chadstone Centre, Victoria, Australia), with RNA sequencing libraries then constructed with the SMARTer Stranded total RNA-seq kit (TaKaRa, Clayton, Victoria, Australia). RNA sequencing of 100 bp pair-end libraries on the Illumina NovaSeq platform yielded 84.54 Gb of data (Appendix A). All sequencing reads have been uploaded onto the NCBI Sequence Read Archive (SRA) database under BioProject PRJNA552109.

RNA sequencing reads were quality trimmed and horse reads were subsequently removed by mapping to the horse genome. To identify potential viral transcripts, non-horse reads from each library were compared against the non-redundant nucleotide (nt) and non-redundant protein (nr) databases using Blastn and Diamond blastx, respectively, with e-value thresholds of 1 × 10^−10^ and 1 × 10^−4^ [10], and were then annotated by taxonomy. Reads from the virus-positive library were *de novo* assembled using Megahit v1.1 [11,12]. Virus-associated contigs were extracted and assembled using Geneious 11.1.5 [13], followed by subsequent blast analysis against the NCBI nt database using BLASTn as further confirmation.

## 3. Results and Discussion

### 3.1. Identification of a Novel Equine Papillomavirus

A 7605 bp genome sequence of a papilloma-like virus was identified in one semen library. Prediction of open reading frames (ORFs) was performed using the ORF Finder tool at NCBI (https://www.ncbi.nlm.nih.gov/orffinder/). A conserved domain search (https://pave.niaid.nih.gov/#analyze/l1_taxonomy_tool) revealed that the L1 protein of the new virus exhibited the highest nucleotide and amino acid identities with EcPV2, at 69.35% and 70.44%, respectively, indicative of a novel papillomavirus. A novel *Equus caballus papillomavirus 8* (EcPV8) associated with viral plaques, viral papillomas, and squamous cell carcinoma has been recently described [14]. We therefore refer to the novel equine papillomavirus described here as *Equus caballus papillomavirus 9* (EcPV9, GenBank accession number MN117918), in accordance with current guidelines for the classification of papillomaviruses [15]. To obtain the full virus genome and to verify the sequence obtained from the deep sequencing and assembly processes, overlapping primers were designed and nested RT-PCR was performed. This resulted in the determination of a circular genome of 7656 bp in length. Remapping of the sequence reads from this library revealed a maximum coverage of 3419X (Figure 1), corresponding to an abundance of 152.97 RPM (reads mapped per million input reads).

### 3.2. Genomic Properties and Evolutionary Relationships of EcPV9

The genome of EcPV9 has a GC content of 52.9% and the classic papillomavirus ORFs were identified, encoding five early (E1, E2, E4, E6, and E7) and two late (L1 and L2) proteins, consistent with other PVs (Figure 2). The predicted nucleotide and amino acid features are summarized in Figure 2 and Appendix A. The noncoding region (NCR) framed by the L1 stop and the E6 start codons comprised 680 nt and exhibited one polyadenylation site (AATAAA) at nt 34. One E1 (TAGATCATTGTTAACAAC) (nt 580), two SP1 (GGCGGG) (nt 5021 and 5084), and three NF1 (CGGAA) (nt 2247, 3144 and 5728) binding sites were also predicted, although the AP1 (TGANTCA) binding site is absent (Figure 2; Appendix A). In addition, 16 typical E2 binding sites were identified comprising 8 true consensus sequences (ACC-N6-GGT) and 8 putative consensus sequences (2 ACC-N4-GGT, 4 ACC-N5-GGT, 2 ACC-N7-GGT) (Figure 2; Appendix A). Such A/T rich N regions are commonplace in E2.

Two zinc-binding domain(s) (CXXC-X29-CXXC) were found in E6 (nt 708 and 937; amino acids 10 and 85) and one in E7 (nt 1201; aa 50), separated by 29 amino acids (Appendix A). No PDZ binding domain (XS/TXV/L) was located at the C-terminus of the predicted EcPV9 E6 protein sequence (Appendix A), which has been reported as a characteristic feature of high risk (i.e., pathogenic) HPV types in comparison to low risk HPVs [16]. Notably, it was previously reported that a PDZ binding domain (XS/TXV/L) was located at the C-terminus of the predicted EcPV-2 E6 protein sequence [8], which was not observed here (Appendix A). No putative pRB binding site (retinoblastoma tumor suppressor-binding domain) (LXCXE) was identified in the putative EcPV9 E7 protein, consistent with all equine and dyoiotapapillomaviruses determined to date [17,18], and the putative E4 protein showed a typical high proline content (12.8%, 18P/141 aa).

We next performed pairwise alignments based on both nucleotide (nt) and amino acid (aa) sequences for the seven viral ORFs. In the case of L1, which is currently used for classification, EcPV9 shared 69.35%, 59.25%, and 58.00% nucleotide similarity with EcPV2, EcPV4, and EcPV5, respectively (Table 1). For E1, EcPV9 shared 62.20%, 51.12%, and 53.81% nucleotide similarity with EcPV2, EcPV4, and EcPV5, respectively. Among the other ORFs, the nucleotide identities with EcPV2, EcPV4, and EcPV5 were between 40.23% and 59.49%, with equivalent amino acid identities between 27.18% and 70.44% (Table 1).

To determine the evolutionary relationships of EcPV9, we inferred a phylogenetic tree based on the concatenated alignment of four coding sequences (E1, E2, L2, and L1). Amino acid sequences (concatenated E1-E2-L2-L1) of 13 equine PVs, as well as the type species of each of the 52 PV genera, were aligned using the E-INS-I algorithm in the MAFFT v7 package [19]. A phylogenetic tree was then estimated using the maximum likelihood method in PhyML 3.0 [20], incorporating the LG+Γ model of amino acid substitution, a SPR branch-swapping algorithm, and 1000 bootstrap replications. This analysis revealed that EcPV9 is clearly related to *Dyoiota* PVs—EcPV2, EcPV4, and EcPV5 (Figure 3). Hence, this evolutionary analysis demonstrates that EcPV9 is a novel species within the genus *Dyoiotapapillomavirus*, yet most closely related to EcPV2, classified as *Dyoiotapapillomavirus 1*.

### 3.3. Disease Association

As no biopsy samples could be taken from this case, it is not possible to confidently determine its significance in the observed pathologies. Nevertheless, the novel EcPV described here was extracted from semen samples, collected when a wart-like lesion was visible on the tip of the penis (Appendix A), and hence compatible with a disease syndrome caused by a papillomavirus. In addition, it was notable that EcPV9 exhibited greatest sequence similarity with EcPV2, a major aetiologic agent of equine squamous cell carcinoma (SCC) disease [8], again compatible with the idea that EcPV9 might also be associated with papillomavirus-related malignancies in horses. Finally, our meta-transcriptomic analysis identified no other likely microbial pathogen in any of the samples analyzed from this stallion.

In conclusion, we report the identification of a novel equine papillomavirus (genus *Dyoiotapapillomavirus*) in a thoroughbred Australian stallion suffering a genital papilloma (“wart”), highlighting the broad diversity of these viruses in horses. Further investigation of the clinical impact of this virus on horse health is clearly merited.

## Figures and Tables

**Figure 1 viruses-11-00713-f001:**
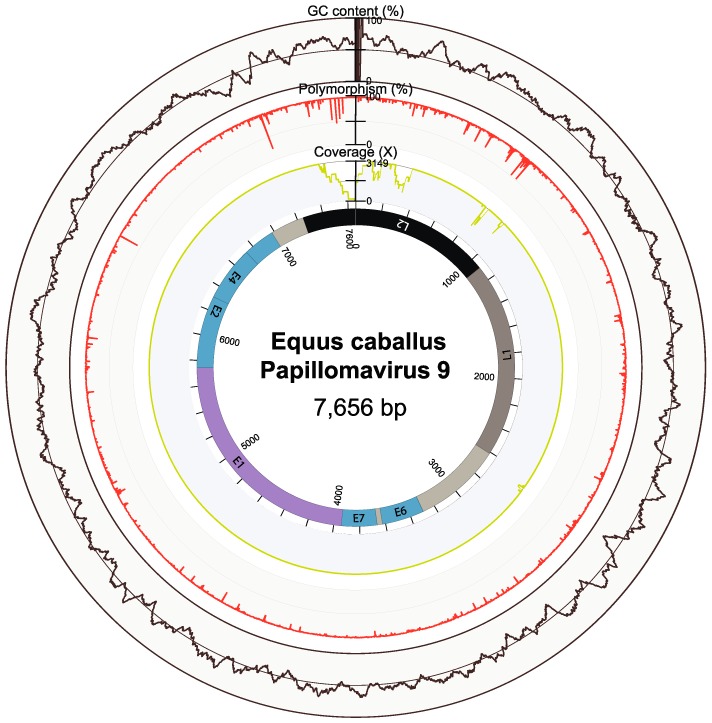
Genome organization of *Equus caballus papillomavirus 9* (EcPV9). The external circles of the metadata ring indicate the percentage GC content (brown), percentage nucleotide polymorphism (red), and read coverage (yellow). The inner gray circle represents the genome, with colored regions showing the predicted open reading frames (ORFs).

**Figure 2 viruses-11-00713-f002:**
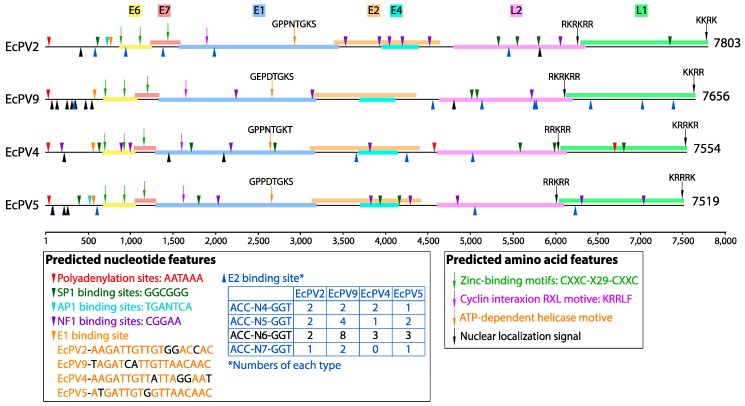
The genome of EcPV9 and three equine papillomaviruses from the genus *Dyoiotapapillomavirus*. Open reading frames (ORFs) are shown at the top and colored according to their putative function. Predicted nucleotide features are shown above and below each genome, using triangles and colored according to different features. Predicted amino acid features are shown above each genome, using down arrows and colored according to the feature in question. Sequences of E1 binding sites are shown in the left box at the bottom with variable nucleotides shown in black type. True consensus E2 binding sites (ACC-N6-GGT) are colored black, while other putative E2 binding sites (ACC-N4-GGT, ACC-N5-GGT, ACC-N7-GGT) are shown in blue, and the numbers of E2 binding sites within each papillomavirus are shown in the left bottom box. Sequences of the ATP-dependent helicase motive and nuclear localization signal are shown above each arrow.

**Figure 3 viruses-11-00713-f003:**
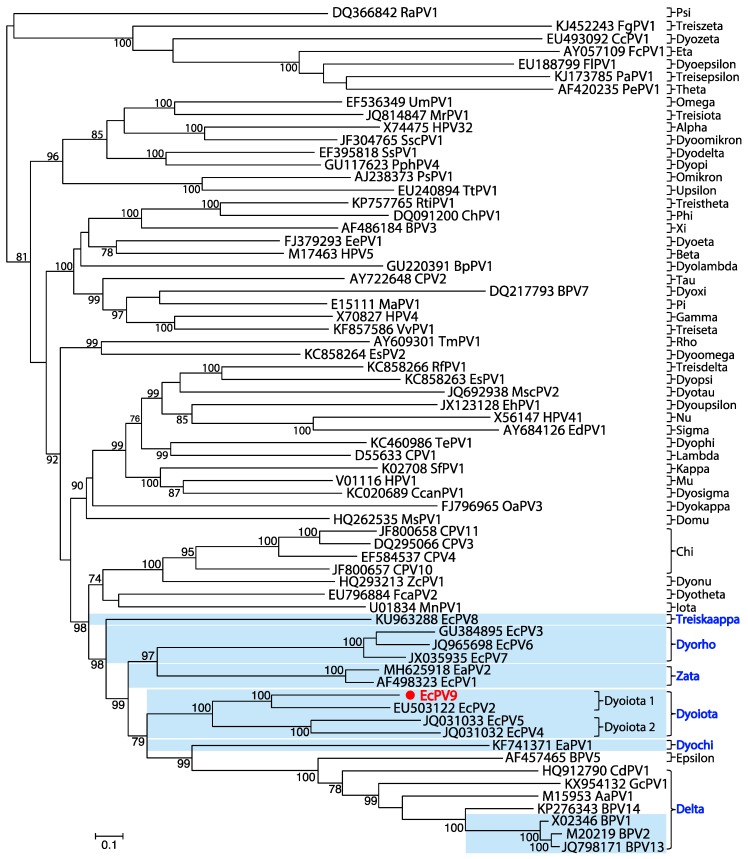
Phylogenetic relationships of EcPV9 to other papillomaviruses based on an analysis of concatenated E1, E2, L2, and L1 amino acids sequences. EcPV9 is shown in red. The names of reference sequences, that contain both the GenBank accession number and the virus species name, are shown in black. Those papillomaviruses associated with horses are shown in a light blue background. The names of previously defined genera are shown to the right of the phylogenies. The tree is mid-point rooted for clarity and nodes supported by >70% of bootstrap replicates are indicated.

**Table 1 viruses-11-00713-t001:** Comparison of sequence similarity based on nucleotide and amino acid sequences of 7 ORFs with EcPV9 to EcPV2, EcPV4, and EcPV5.

EcPV9		EcPV2 (%)	EcPV4 (%)	EcPV5 (%)
E6	nt	52.85	42.93	44.17
	aa	41.86	35.16	34.88
E7	nt	40.23	43.02	50.74
	aa	33.04	27.18	33.98
E1	nt	62.20	51.12	53.81
	aa	58.63	44.97	46.56
E2	nt	51.05	49.54	48.77
	aa	42.79	35.31	36.38
E4	nt	49.32	40.83	42.19
	aa	40.14	33.33	32.24
L2	nt	59.49	46.60	49.63
	aa	59.17	42.55	45.07
L1	nt	69.35	59.25	58.00
	aa	70.44	57.74	59.88

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
