# Peer review of "Identification of a Novel Equine Papillomavirus in Semen from a Thoroughbred Stallion with a Penile Lesion"

_viruses, 2019, doi:10.3390/v11080713_

Round 1

Reviewer 1 Report

The authors describe the isolation of a novel equine papillomavirus.

The article is very well written and it makes a valuable contribution to this scientific field.

However, the title and the conclusion should be modified as this virus cannot be associated to the penile lesion with their data obtained from semen. Only the presence of viral DNA/RNA in the lesion would clearly allow linking this virus to the disease. Also, the authors should explain why the phylogenetic tree has been constructed using concatenated E1-E2-L2-L1 amino acid sequences, instead of L1 only?

Author Response

REVIEWER #1

The authors describe the isolation of a novel equine papillomavirus.

The article is very well written and it makes a valuable contribution to this scientific field.

Response: We thank the reviewer for these positive comments.

However, the title and the conclusion should be modified as this virus cannot be associated to the penile lesion with their data obtained from semen. Only the presence of viral DNA/RNA in the lesion would clearly allow linking this virus to the disease. Also, the authors should explain why the phylogenetic tree has been constructed using concatenated E1-E2-L2-L1 amino acid sequences, instead of L1 only?

Response: We agree that we cannot conclusively associate the virus with the penile lesion (and made this point in the paper), although we believe that the circumstantial evidence is very strong. To help strengthen our case we have added a photo of the lesion as supplementary information, and provided a little more clinical information. As pointed out in response to Reviewer #3, and now in the main text, unfortunately no sample was taken directly from the penile lesion so nothing more can be done to strengthen the link between virus and lesion. However, as requested by the reviewer, we have now revised the title and the concluding paragraph.

We estimated phylogenies on the concatenated E1-E2-L2-L1 amino acid sequences, rather than L1 only, because they obviously contain more evolutionary information (because more amino acid sites are being analysed) and hence will be more accurate.

Reviewer 2 Report

Li et al. describe in their manuscript the detection and genetic characterisation of a not yet described equine papillomavirus. On the penis of a stallion, a wart-like lesion was observed. Total RNA was extracted from semen and urine and next generation sequencing was performed. DNA and protein motifs are identified on the determined viral genome sequence. The sequence was further analysed and compared to other equine papillomavirus sequences. The similarities of the putative coding sequences were calculated and phylogeny based on concatenated protein sequences was inferred. 

The manuscript is clearly written and interesting. However, there are some points, which should be clarified in the text. 

As stated in the Introduction, papillomaviruses are DNA viruses. It does not become clear why the authors performed the sequencing on RNA and not on DNA. Moreover, why is the coverage high over the entire genome except the L2 gene. One would expect a lower coverage over the NCR. The refinement of the nucleotide sequence was performed by nested RT-PCR. Again, the sequence was determined on RNA. There is a difference of 51 bp between the sequences determined by RNAseq and by RT-PCR. What part of the genome is covered by this difference? Since the sequencing was performed on RNA, is there an information about the splicing of the viral mRNA? 

Author Response

REVIEWER #2

Li et al. describe in their manuscript the detection and genetic characterisation of a not yet described equine papillomavirus. On the penis of a stallion, a wart-like lesion was observed. Total RNA was extracted from semen and urine and next generation sequencing was performed. DNA and protein motifs are identified on the determined viral genome sequence. The sequence was further analysed and compared to other equine papillomavirus sequences. The similarities of the putative coding sequences were calculated and phylogeny based on concatenated protein sequences was inferred. 

The manuscript is clearly written and interesting. However, there are some points, which should be clarified in the text. 

As stated in the Introduction, papillomaviruses are DNA viruses. It does not become clear why the authors performed the sequencing on RNA and not on DNA. Moreover, why is the coverage high over the entire genome except the L2 gene. One would expect a lower coverage over the NCR. The refinement of the nucleotide sequence was performed by nested RT-PCR. Again, the sequence was determined on RNA. There is a difference of 51 bp between the sequences determined by RNAseq and by RT-PCR. What part of the genome is covered by this difference? Since the sequencing was performed on RNA, is there an information about the splicing of the viral mRNA? 

Response: Bulk RNA-sequencing ("meta-transcriptomics") is our preferred method of pathogen discovery because it is able to identify *any* type of pathogen that has expressed RNA at the time of sampling (RNA viruses, DNA viruses, bacteria, fungi, parasites). At the time of the study the pathogen causing this disease was unknown, so we used RNA-sequencing to give us the best chance of finding it: clearly this worked! If, on the other hand, the lesion had been caused by an RNA virus we would have missed it with only DNA sequencing. We have now better outlined the rationale behind our approach in the revised version of the paper.

There are two possible reasons for the lower coverage of the L2 gene: that it is expressed at lower levels or that it simply reflects random sampling. Unfortunately, it is not possible to determine which is correct. The 51 bp gap was in the middle of L2 gene and was completed by RT-PCR. Finally, because the complete virus genome was verified by overlapping primers and RT-PCR, we unfortunately did not recover any information on viral mRNA splicing.

Reviewer 3 Report

This short manuscript describes the identification of a novel equine papillomavirus that they have called EcPV9. Its closest homolog is EcPV2 which is a causative agent in penile cancer in horses. Therefore, this virus could be involved in the causation of equine disease. 

Why did the authors not take a sample from the actual genital lesion and assay it? They state no biopsy could be taken but surely a swab of the lesion could have been taken? Is it the same virus they detect in the urine and semen? Nothing they can do about this now but it should be commented on. A photograph of the lesion would have been helpful for others in the field, if they have this I encourage them to include it in the manuscript.

There are many predicted E2 DNA binding sites. The true consensus for E2 is ACCN6GGT, it would be useful for the authors to highlight these as distinct from their other putative E2 binding sites. In addition, A/T rich N regions are favored by E2 and this could also be commented on.

Author Response

REVIEWER #3

This short manuscript describes the identification of a novel equine papillomavirus that they have called EcPV9. Its closest homolog is EcPV2 which is a causative agent in penile cancer in horses. Therefore, this virus could be involved in the causation of equine disease. 

Why did the authors not take a sample from the actual genital lesion and assay it? They state no biopsy could be taken but surely a swab of the lesion could have been taken? Is it the same virus they detect in the urine and semen? Nothing they can do about this now but it should be commented on. A photograph of the lesion would have been helpful for others in the field, if they have this I encourage them to include it in the manuscript.

Response: Unfortunately, no sample was taken from the genital lesion so there is nothing more we can do here. We are as frustrated as the reviewer, but our hands are tied. We have now made this very clear in the text. However, as requested, we have now added a photograph of the penile lesion as supplementary information: as the picture is rather low resolution and is perhaps not entirely appropriate for a mainstream virology journal, we thought it was best suited to the supplementary information.

There are many predicted E2 DNA binding sites. The true consensus for E2 is ACCN6GGT, it would be useful for the authors to highlight these as distinct from their other putative E2 binding sites. In addition, A/T rich N regions are favoured by E2 and this could also be commented on.

Response: We thank the author for this information. We have revised our description of the E2 binding sites accordingly - in the main text, in Figure 2, and in the legend to this figure.

Round 2

Reviewer 3 Report

The authors have adequately responded to the issues raised. I'd like to point out a couple of things.

It's fine to justify RNA-seq as a mode of discovery and it is worth pointing out to the reviewer that DNA-seq is much more expensive and much more difficult to analyze.

The reason there is no L2 expression (raised by another reviewer) is likely because the virus detected is not from differentiated cells. In the HPV life cycle the L2 RNA and protein is only detected in terminally differentiated cells (where it encapsulates the viral DNA genome to form viral particles that egress from the "top" of the differentiated epithelium), therefore it is not surprising that they have not detected the L2 RNA in their samples as equine PVs probably follow a similar pattern of gene regulation during differentiation. It is certainly worth pointing this out in the manuscript.